

# ELIFAN, an algorithm for the estimation of cloud cover from sky imagers

Marie LOTHON[1], Paul BARNÉOUD[1], Omar GABELLA[1], Fabienne LOHOU[1], Solène DERRIEN[1],
Sylvain RONDI[2], Marjolaine CHIRIACO[3], Sophie BASTIN[3], Jean-Charles DUPONT[4],
Martial HAEFFELIN[5], Jordi BADOSA[6], Nicolas PASCAL[7], and Nadège MONTOUX[8]

[1]Laboratoire d'Aérologie, Université de Toulouse, CNRS, UPS, France
[2]Direction Académique Hautes-Pyrénées, Tarbes, France; formerly Laboratoire d'Astrophysique, Toulouse-Tarbes, France
[3]LATMOS/IPSL, UVSQ Université Paris-Saclay, Sorbonne Université, CNRS, Guyancourt, France
[4]Institut Pierre-Simon Laplace, École Polytechnique, UVSQ, Université Paris-Saclay, 91128 Palaiseau, France
[5]Institut Pierre-Simon Laplace, École Polytechnique, CNRS, Université Paris-Saclay, 91128 Palaiseau, France
[6]LMD, IPSL, École Polytechnique, Université Paris-Saclay, ENS PSL Research University, Sorbonne Université, CNRS, 91128 Palaiseau France
[7]ICARE, Lilles, France
[8]Université Clermont Auvergne, CNRS, LaMP, F-63000 Clermont-Ferrand

*Correspondence to:* Marie LOTHON (marie.lothon@aero.obs-mip.fr)

**Abstract.**

In the context of an atmospheric network of multi-instrumented sites equipped with sky camera for cloud monitoring, we present an algorithm named ELIFAN which aims at estimating the cloud cover amount from full sky visible daytime images with a common principle and procedure. ELIFAN was initially developped for a self-made full sky image system presented in this article, and adapted to a set of other systems in the network. It is based on red over blue ratio thresholding for the distinction of cloudy and clear sky pixels of the image, and on the use of a blue sky library. Both an absolute (without use of reference image) and a differential (based on a blue sky reference image) red/blue ratio thresholding are used.

An evaluation of the algorithm based on a one-year long series of images shows that the proposed algorithm is very convincing for most of the images, with more than 95% of relevance in the process, outside the sunrise and sunset transitions. During those latter periods though, ELIFAN has large difficulties to appropriately process the image, due to a drastic difference of color composition and a potential confusion between clear and cloudy sky at that time.

The two thresholding methodologies, the absolute and the differential red/blue ratio thresholding processes, agree very well with departure usually below 8%, except in sunrise/sunset periods and in some specific conditions. The use of the clear sky library gives generally better results than the absolute process. Especially, it better detects the thin cirrus clouds. But the absolute thresholding process turns out to be sometimes better, for example in fully cloudy skies.

The combination of pyranometer, ceilometer and sky camera illustrates the performance of ELIFAN, and reveals the complementarity of the three instruments. We especially show that a similar cloud cover amount is deduced from both the sky imager and the ceilometer when the clouds are low (below 3 km). But they can lead to significantly different cloud cover estimates



when the clouds are high. In this case, we find that the sky imager catches more appropriately the cloud cover estimate, due to its 2D integrated point of view and to the loss of sensitivity of the ceilometer above 7 km.

**Keywords:** Sky imager, Cloud cover, Cloud fraction, Algorithm, Image processing, Instrumented Sites, ACTRIS-FR



# 1 Introduction

Due to their crucial role in weather and climate, clouds are the focus of many observation systems all over the world. Sky imagers are naturally used as simple devices for visible sky monitoring: They give a very useful qualitative information of the state of the sky and of the type of clouds. But they can also fulfill quantitative parameters, after a process of the image that is either based on the texture of the image, or on the {Red, Green, Blue} color composition of the image. Several algorithms have now been proposed, which enable to retrieve an estimation of cloud cover (e. g. Long et al. (2006), Li et al. (2011), Ghonima et al. (2012), Martinis et al. (2013), Silva and Echer (2013), Cazorla et al. (2015), Kim et al. (2016), Krinitskiy and Sinitsyn (2016)) or solar irradiance (Pfister et al. (2003), Chu et al. (2014), Chauvin et al. (2015), Kurtz and Kleissl (2017)), to classify the type of observed clouds (e. g. Heinle et al. (2010), Kazantzidis et al. (2012), Xia et al. (2015), Gan et al. (2017)), or to track them (Peng et al. (2015), Cheng (2017), Richardson et al. (2017)). Sky imagers have also been specifically used for the detection of cirrus (Yang et al., 2012) or thin clouds (Li et al., 2012), and contrail studies (Schumann et al., 2013). Furthermore, one can also estimate the cloud base height (Allmen and Kegelmeyer (1996), Kassianov et al. (2005), Nguyen and Kleissl (2014)) by using a pair of sky cameras.

Those systems are now commonly deployed in the vicinity of solar farms for the intra-hour or now-casting of solar irradiance, and during atmospheric field experiments or on permanent observatories, for the cloud cover and cloud type monitoring.

Within the ACTRIS-FR [1] French research infrastructure, a network of instrumented permanent sites has coordinated their actions for the observation of the atmosphere, and attempts to homogenize their instrumental, data process, and data dissemination practises, for a larger and more consistent multi-parameter data use of the international research community. In this context, a common sky imager algorithm has been developped, called ELIFAN, in order to retrieve in a similar way the cloud fraction from all the sky cameras of the network. It is now used on three ACTRIS-FR sites, and in progress on three other sites. ELIFAN is based on red over blue ratio (here after RBR) thresholding for the distinction of cloudy and clear sky pixels of the image, and on the use of a blue sky library. This article aims at presenting ELIFAN algorithm principle, strength, weaknesses and perspectives.

In section 2, we present the sky cameras used in the ACTRIS-FR infrastructure, with more details on a self-made sky imager developed at one of the instrumented sites, and on which ELIFAN was originally based. In section 3, ELIFAN algorithm is explained in details, with highlights on its main strengths and on its weaknesses. In section 4, we illustrate it with the analysis of two days, where the cloud cover estimated by ELIFAN is compared with ceilometer and pyranometer measurements. We make concluding remarks in the last section, with perspectives of evolution of the algorithm and further discussion.

---

[1]ACTRIS-FR is the French component of the European Aerosol, Cloud and Trace Gases Research Infrastructure (ACTRIS), http://cache.media.enseignementsup-recherche.gouv.fr/file/Infrastructures_de_recherche/70/3/Brochure_Infrastructures_2018_948703.pdf



## 2 Sky Imager systems used

### 2.1 The Sky Imager systems of ACTRIS-FR Instrumented Sites

There are five important multi-instrumented sites which participate to the French infrastructurue ACTRIS-FR, and are spread over the French territory:

- P2OA (Pyrenean Platform for the Observation of the Atmosphere[2]), in the Pyrénées, near the Spanish border, in Southwest France, with two sites: one at Pic du Midi summit (42.94°N, 0.143°E) and the other close to Lannemezan, at the 'Centre de Recherches Atmosphériques' (43.13°N, 0.366°E),

- SIRTA ('Site Instrumental de Recherche par Télédétection Atmosphérique', Haeffelin et al. (2005)), at Palaiseau, south of Paris (48.72°N, 2.21°E),

- CO-PDD ('Cézeaux –Opme- Puy de Dôme'[3]) in center of France, with 3 sites: one at the Puy de Dôme summit (1465 m, 45.77°N, 2.96°E), one at Opme (680 m, 45.71°N, 3.09°E) and one at the Cézeaux University site (410 m, 45.76°N, 3.11° E)

- OHP ('Observatoire de Haute Provence'[4]) in the Provence region, in Southeast France (43.93°N, 5.71°E),

- OPAR ('Observatoire de Physique de l'Atmosphère de la Réunion'[5]) at the Reunion Island (21.08°N, 55.38°E).

15 All have a sky camera for the cloud monitoring, but different systems have been historically used: TSI (Total Sky Imager) (used at SIRTA, from 23/10/2008 to 24/06/2015), ASI (All Sky Imager) from EKO (used at SIRTA, CO-PDD and P2OA mountain site), Alcor System (used at OPAR), and self-made instruments (RAPACE, used at P2OA plain site, and another one at OHP).

Table 1 summarizes the systems used at 3 observatories, for which the sky images are currently processed for the cloud cover 20 retrieval.

**Table 1.** List of sky imager systems within the ACTRIS-FR network and their characteristics. The list here only consider the systems for which the images are processed by ELIFAN for cloud cover retrieval.

| Site | Sky imager system | Solar mask | Image size | Time interval | Start year | Comment |
|------|-------------------|------------|------------|---------------|------------|---------|
| P2OA | RAPACE | no | 2048 x 1536 | 5 min | 2006 | at plain site |
| P2OA | EKO - SRF02 | no | 2272 x 1704 | 5 min | 2017 | at mountain site |
| SIRTA | TSI - 440 | yes | 640 x 480 | 1 min | 2008 | peri-urban site (former system) |
| SIRTA | EKO - SRF02 | no | 1024 x 768 | 2 min | 2014 | peri-urban site (current system) |
| CO-PDD | EKO - SRF02 | no | 1024 x 768 | 1 min | 2015 | at plain site |

[2]http://p2oa.aero.obs-mip.fr/
[3]http://www.opgc.univ-bpclermont.fr/SO/mesures/
[4]http://www.obs-hp.fr/welcome.shtml
[5]http://lacy.univ-reunion.fr/observations/observatoire-du-maido-opar/





Initially, the image processing algorithm was developped and profoundly evalutated on RAPACE system images. But it was subsequently adapted for the other commercial systems of the network, in order to realize a common data process within ACTRIS-FR sky imager network. The network also currently tends to homogenize the sky imager systems.

In the following subsection, we describe in more details the RAPACE self-made system that is at the origin of the ELIFAN
algorithm developement, and which will be considered for the illustrations of further sections.

## 2.2 RAPACE system

There are several self-made systems presented in the litterature that show that this is often a quite satisfactory option when trying to acquire good quality images at significantly lower cost than commercial systems. The developers then also design their own specific data process, depending on their objectives. Kazantzidis et al. (2012) have made a whole sky imaging system
based on a commercial digital camera with a fish-eye lens and a hemispheric dome, for the automatic estimation of total cloud coverage and classification. Cazorla et al. (2008) have used a CCD camera for the purpose of cloud cover estimation and characterisation. The system captures multispectral image every 5 min. Chu et al. (2014) has proposed an automatic smart adaptative cloud identification (SACI) system for sky imagery and solar irradiance forecast, which uses an off-the-shelf fish-eye. Jayadevan et al. (2015) also uses a fishe-eye, for their sun-tracking camera. Urquhart et al. (2015) have developed a high
dynamic range (HDR) camera system capable of providing a full sky multispectral image at radiometric resolution, every 1.3 s.

The instrument used at P2OA plain site, called RAPACE ('Récepteur Automatique Pour l'Acquisition du Ciel Entier') was made in December 2006, based on the same very simple principle as Kazantzidis et al. (2012), in the purpose of taking automatic and regular whole sky images.
RAPACE System is composed of:

- a A510 CANON camera, remotely controllable,

- a FC-E9 fish-eye objective (from Edmond Optics) , for full sky image

- a waterproof box, for protection of the camera outside,

- a square board support, to put the camera at the right height into the box,

- a Plexiglas dome, to protect the fish-eye objective from hail, animals, and other sources of damage,

- a thermostatted heating wire, to limit condensation and frost inside the dome,

- a control computer,

- Power Shot Remote software (from Breeze Systems), for remote control of the camera.

A picture of RAPACE is shown in Fig. 1, on the roof of the P2OA plain site laboratory building. Exemples of images are
shown in Fig. 2.



It has now been running since December 2006 with no significant interruption, and turned out to be a very robust system, with high quality 3.2 M pixels images taken every 15 min until February 2017 and every 5 min (during daytime, 15 min during nighttime) since then.

Assuming that the main fragility would lie in the mechanical constraint endured during (i) the successive opening and closing
5 of the digital camera objective, and (ii) focusing, we have blocked the objective in opened position, and also fixed the focus after focusing to infinity. This seemed to help a lot in the system endurance, as RAPACE has run for 12.5 years now with the same original digital camera and fish-eye. Only the Plexiglas dome has been replaced a couple of times due to hail damage, and the USB extension wires have been improved along the time.

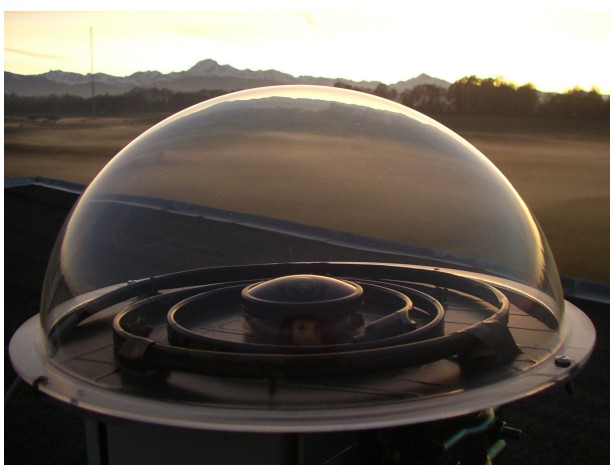

**Figure 1.** The RAPACE sky imager system developed at P2OA instrumented site. We can see the fish-eye at the center, the Plexiglas dome and the thermostatted heating wire in spiral around the fish eye.

(a)                                         (b)

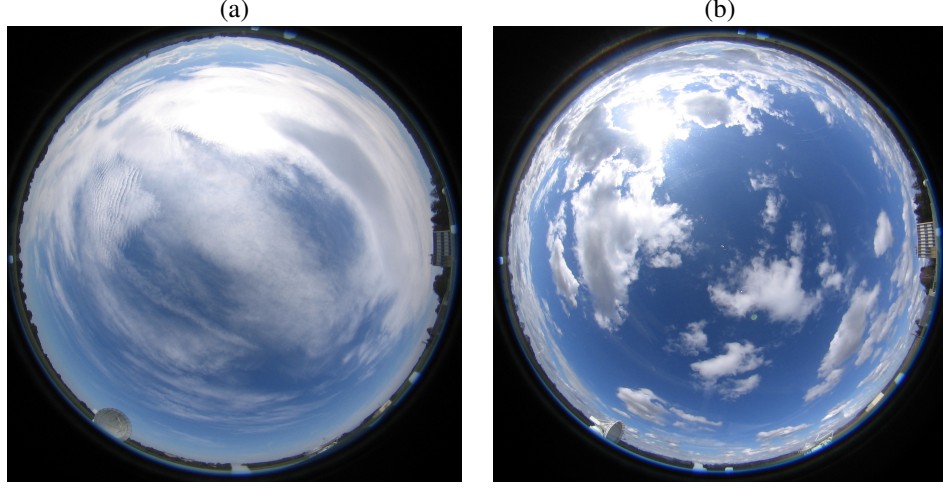

**Figure 2.** Two examples of RAPACE full sky images. (a) Cirrus clouds on 2 October 2007, (b) Cumulus clouds on 20 February 2006.



In the purpose of using a common and centralised data process to retrieve the cloud cover from the images of all the sky cameras available at the instrumented sites of the ACTRIS-FR network, an algorithm has been developed. It is now operationnally running at AERIS (Data and Services for the atmosphere) - ICARE (Data and Services Center) data center for the images of all the EKO systems listed in Tab. 1, and of RAPACE. This algorithm is presented in details in the next section.

## 3 ELIFAN algorithm

### 3.1 Background on the retrieval of cloud cover from a sky camera

They are several ways to detect clouds and estimate the cloud cover percentage from a visible full sky image. Table 2 summaries the different methodologies found in the litterature. Most of the algorithms lean on the colour information of the image pixels, but there are also techniques based on the texture of the image. Combining both increase the retrieval capacity, and especially gives more possibility for cloud types identification.

Within the first category, one simple and though quite efficient way to proceed is based on the use of thresholds on the RBR (Long et al., 2006). This methodology is based on the fact that a blue sky is characterised by pixels with larger contribution of blue relative to the two other colours, while clouds have more homogeneous relative contributions (closer to white or gray). The methodology can be sophisticated with combination of RBR and GBR (Green over Blue ratio) (Kim et al., 2016), or the use of saturation (Souza-Echer et al., 2006). One of the main difficulties lies in the variability of the colour of the clear sky, from one day to another (for example due to variable aerosol loading and hydratation), and from one time to the other (due to variable sun elevation). The use of adaptative thresholding (Li et al. (2011), Jayadevan et al. (2015)) helps improving the results on this aspect. Another permanent challenge is due to the variability of the colour of clear sky from one part of the image to the other. There is indeed heterogeneity within a totally clear sky image, due to forward scattering and Mie scattering of aerosols. Depending on the position of the pixel relatively to the sun position, the blue sky appears differently. It especially makes the circumsolar area very difficult to deal with (there is an increase whitening around the sun). To solve this issue, Ghonima et al. (2012) uses a real clear sky library as a reference, composed of a set of clear sky images found within a large dataset. The difference between the RBR of the processed image and the RBR of a reference clear sky image (with similar elevation and azimuth solar angles) departs the cloudy pixels from the clear pixels. Going further, recent methodologies based on background substraction take account of the blue sky spatial and temporal variability, by use of a modeled (or so-called 'virtual') clear sky (Yabuki et al. (2014), Chauvin et al. (2015), Yang et al. (2016)).

Mathematical tools like neural network, K-nearest neighbour techniques, or Support Vector Machine (SVM) are also used for cloud detection, based on the RGB composition and luminosity (Blazek and Pata (2015), Taravat et al. (2015), Cheng (2017)), and on the image texture (Cazorla et al. (2008), Kazantzidis et al. (2012), Liu et al. (2015), Peng et al. (2015)). The combination of both allowed Kazantzidis et al. (2012) to distinguish seven types of clouds, in addition to estimating cloud cover amount. Roman et al. (2017) interestingly uses the fact that an image of totally clear sky is symmetric.



**Table 2.** Synthetic table of background studies of cloud cover retrieval algorithms from sky camera.

| Feature | Basic principle | Diagnostic | Reference |
|---|---|---|---|
| Colour | Thresholding | RBR | Long et al. (2006) |
| Colour | Thresholding | RBR, GBR | Kim et al. (2016) |
| Colour | Thresholding | Saturation | Souza-Echer et al. (2006) |
| Colour | Adaptative Thresholding | normalised BRR | Li et al. (2011) |
| Colour | Adaptative Thresholding, contrast enhancing | normalised saturation value ratio | Jayadevan et al. (2015) |
| Colour | Adaptative Thresholding, Background substraction | G | Yang et al. (2015) |
| Colour | Clear sky library, Thresholding | RBR | Ghonima et al. (2012) |
| Colour | Clear sky modeling, Background substraction | normalised RBR | Chauvin et al. (2015) |
| Colour | Clear sky modeling, Background substraction | G | Yang et al. (2016) |
| Colour | Clear sky modeling, Background substraction | B, G, BRR | Yabuki et al. (2014) |
| Colour | Adaptative filtering of Mie Scattering contribution | Grayness rate index | Krinitskiy and Sinitsyn (2016) |
| Colour | Colour transformation, K-means segmentation | RGB and luminosity | Blazek and Pata (2015) |
| Colour | Neural network | RGB | Taravat et al. (2015) |
| Colour | Neural network, neighborhood | R, G, B, all ratios | Cazorla et al. (2008) |
| Colour | Multiscale neighborhood and multiple learning | RGB, HSV, YCbCr | Cheng (2017) |
| Colour and Texture | Superpixel segmentation, thresholding | R-B | Liu et al. (2015) |
| Colour and Texture | Symetry of sky | R, B | Roman et al. (2017) |
| Colour and Texture | Neighborhood, Support Vector Machine | R, G, B, RBR, luminance | Peng et al. (2015) |
| Color and Texture | K-nearest neighbour | R, B | Kazantzidis et al. (2012) |

Finally, there is of course a gain in combining instruments. Ceilometers and sky imagers are very complementary in this prospect, as the ceilometer adds the cloud base height information above the sky camera system (Chu et al. (2014), Roman et al. (2017)), which can allow to access to cloud speed motion (Wang et al., 2016). Pyranometers are also naturally considered in such instrumental synergy (Peng et al., 2015).

## 3.2 Principle and methodology of ELIFAN algorithm

Initially developed in 2013, ELIFAN algorithm aims at estimating the cloud cover percentage during daytime, based on a visible image. At that time, only RAPACE system at P2OA site and the TSI-440 system of SIRTA site were used within the French network of instrumented sites. An algorithm that could be used for both cameras, despithe their differences, needed to be developed. ELIFAN is basically inspired by Ghonima et al. (2012), with the use of the RBR as the driving diagnostic to depart cloudy and clear air pixels by thresholding, and of a reference blue sky library. Both an absolute and a differential thresholding processes are applied.

### 3.2.1 The different steps along the process

For a given image to be processed, the different steps are the followings:

– Step1: The image is cropped in order to remove the obstacles at horizon and the circumferential part of the image that is too distorted. Figure 3 shows an example of a raw RAPACE image (26 February 2014 at 1300 UTC) and the





corresponding cropped selection (step 1). Only the pixels located into the cropped image (i. e. inside the black circle in figure 3a) are processed.

– Step 2: A Solar mask is applied, and other masks if needed (e. g. for TSI camera, which has a sun mask arm). The solar maks is positioned with a corrected function of elevation and azimuth angles, which are deduced from the localisation of the site, date and time.

– Step 3 (see section 3.2.2): With a combination of criteria applied on the global probability density function (pdf) of the image RBR, a primary phase evaluates whether the image is totally cloudy, totally clear or partly cloudy:

  – A totally cloudy image is associated with 100% cloud cover;

  – A totally clear image is sent to the clear sky library, and associated with a 0% cloud cover;

  – A partly cloudy image continues the process with the following step.

– Step 4 (see section 3.2.3): As a secondary phase, all images considered as partly cloudy image during the previous step 3 are processed, with a pixel by pixel point of view. For this, the algorithm searches for a reference blue sky image within a library, with the sun at same azimut and same elevation, $\pm 1°$, as the considered image.

  – If there is no reference image, the image is processed with absolute RBR threshold process

  – If there is a reference image, the image is processed with both the absolute RBR threshold process and the differential RBR threshold process. If there are several reference images available, the sky with the least turbidity is chosen as a reference, based on the RBR pdf.

Cloud cover estimate is given by the percentage of cloudy pixels / uncertain pixels / blue sky pixels.

(a)             (b)

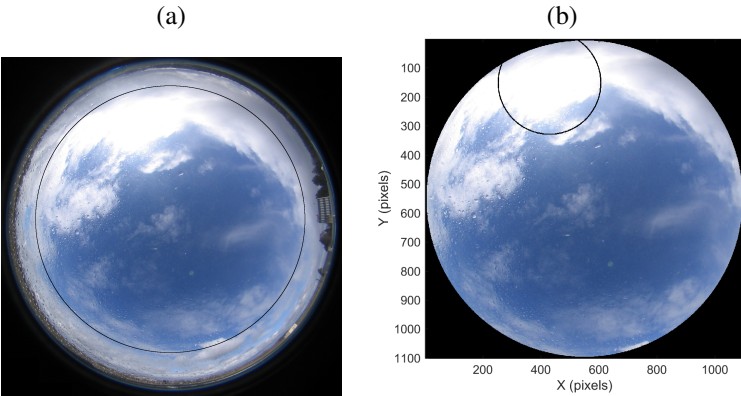

**Figure 3.** (a) Initial image 26.02.2018 at 1300 UTC. (b) Corresponding cropped image (Step 1 of the process). In (a), the black circles indicates the contour of the cropped image shown in (b), outside which the pixels are not processed. In (b), the black circle indicates the contour of the sun mask, within which the pixels are not processed.

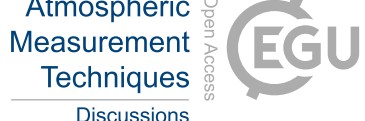



A scheme of the pathway followed by the image and summarizing those steps is drawn in Fig. 4, and steps 3 and 4 mentioned above are explained and illustrated in more details in the following subsections.

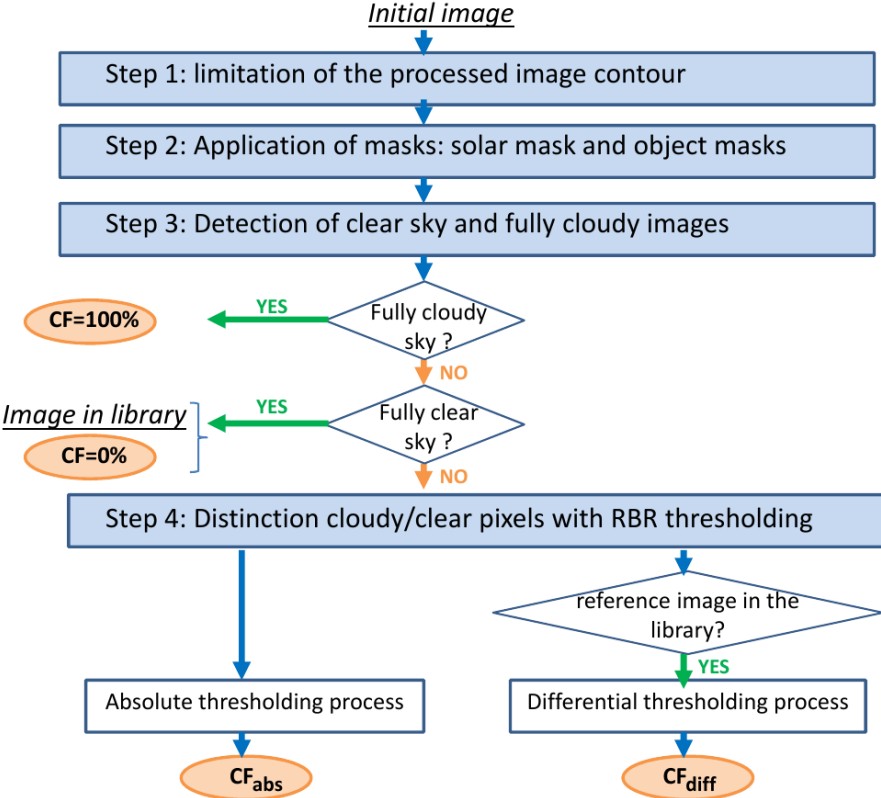

**Figure 4.** Organigram of the various steps followed along the image processing. $CF_{abs}$ is the cloud fraction deduced from the absolute thresholding process, and $CF_{diff}$ the cloud fraction deduced from the differential thresholding process.





### 3.2.2   Step 3: Detection of clear sky and fully cloudy sky images

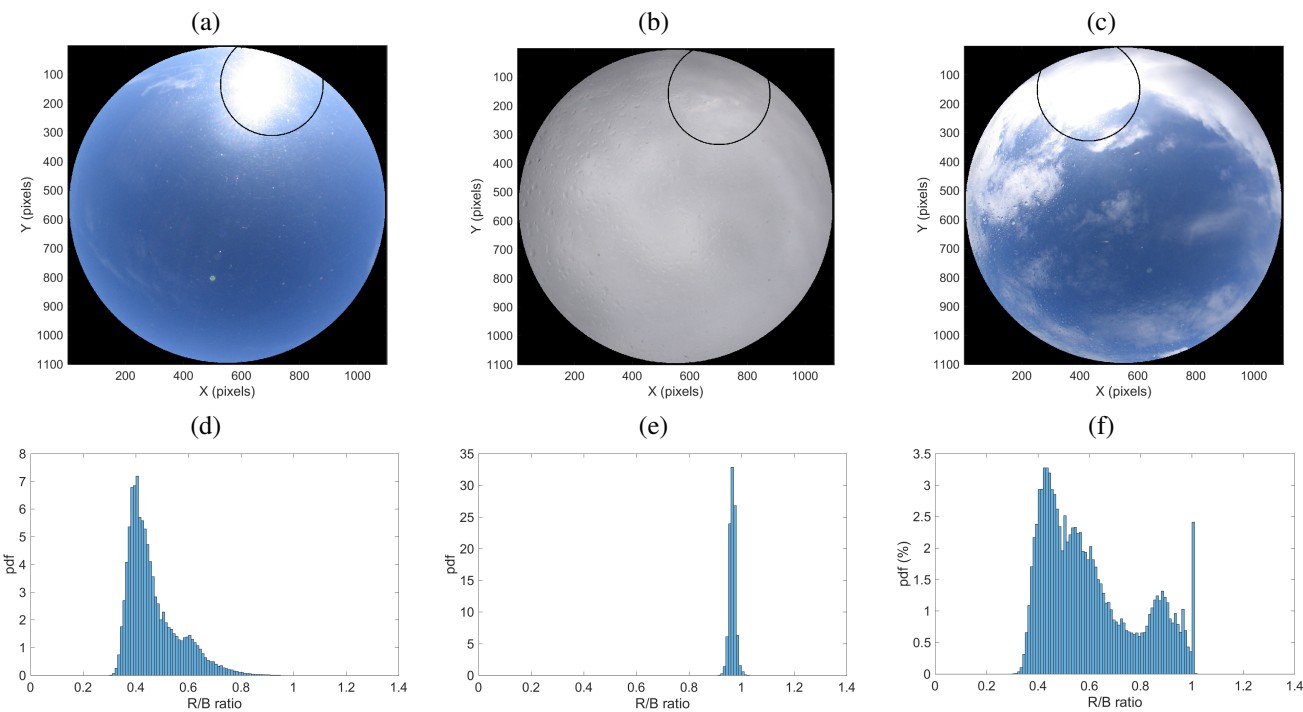

**Figure 5.** Cropped images of (a) 17.02.2018 at 1100 UTC, (b) 26.02.2018 at 1100 UTC, (c) 26.02.2018 at 1300 UTC, and (d, e, f) the corresponding pdf of the RBR respectively.

In step 3, the principle is to consider the cropped image as a whole, and detect whether it is a fully cloudy sky image, a full clear sky image, or neither of those two (i. e. a partly cloudy image). This step is based on the RBR distribution of the entire ensemble of pixels.

Figure 5 shows three examples: an image with clear sky (Fig. 5a), a second with fully cloudy sky (Fig. 5b) and a third with partly cloudy sky (Fig. 5c, which is the same as in Fig. 3). In the first image (clear sky, Fig. 5a), all the pixels have RBR < 0.75. The spread from 0.3 to 0.8 is due to the variability of the RBR with the scattering angle from zenith. Ghonima et al. (2012) have shown how the RBR in a given circular band of a clear sky image also depends on the aerosol optical depth (AOD), varying almost linearly with AOD from 0.3 to 0.7 in their analysis for AOD within [0, 0.3], for scattering (zenith) angle within

[0.35°-0.45°].

In the second image (fully cloudy, Fig. 5b), most of the pixels have RBR > 0.75. Thicker clouds have RBR closer to 1 (Ghonima et al., 2012).

In the third image (partly cloudy, Fig. 5c) the probability density function (pdf) of the RBR shows a bimodal distribution, corresponding to the two sets of clear sky pixels in one hand (left side mode) and cloudy sky pixels in the other hand (right

side mode).





The main goal of step 3 is to detect whether the image is a clear sky image, to be transfered to the library. It also detects whether the image is a fully cloudy image (CF=100%).

If the maximum of RBR over the entire image is smaller than 0.6 or if more than 98% of the pixels have RBR > 0.75 (like in Fig. 5b), then this image is defined as a fully cloudy image.

Otherwise, we check whether the image has less than 2.5% of the pixels with RBR > 0.75, but still a few pixels with RBR > 0.85. A clear sky image meets this criterium, because it has most of its pixels with RBR < 0.75 (see Fig. 5a), but due to a few white pixels in the circumsolar region (around the sun mask here), there will be (only) a few pixels with very high RBR (> 0.85).

So we then verify whether the image has some cloudy pixels, or is really clear sky. If it meets at least 2 criteria among the
following three criteria, it means that there are clouds in the image:

– More than 90% of the pixels have RBR within [0.45, 0.65];

– The maximum probability within the [0.45, 0.65] range is larger than 35% ;

– Less than 12.5% of the pixels have RBR $\leq$ 0.5.

Any image detected as full clear sky image after this test (that is no cloud has been detected with the previous test) is sent
to the clear sky library, and associated with the azimuth and solar zenith angle corresponding of the site and time of the image. That is how the reference library is progressively built.

At this point, any image which was not detected as clear sky image, or as a fully cloudy sky image, will be going through the process of step 4, for the estimate of the cloud fraction.

Note that the combination of the criteria explained above varies according to the sky camera. But they are all based on the
RBR distribution over the entire set of pixels, and on the same main principle.

### 3.2.3    Step 4: Distinction of cloudy and clear sky pixels in a partly cloudy image

In step 4, the considered image, which is partly cloudy by construction, is now considered at the pixel point of view, —i. e. processed pixel by pixel—, contrarily to step 3. It is independently submitted to both an absolute thresholding process and a differential thresholding process when a reference clear sky image exists:

– The absolute thresholding process compares the RBR of each pixel to 2 thresholds $T_{clear}$ and $T_{cloud}$:

If $RBR \leq T_{clear}$, the pixel is considered as 'blue',

if $RBR \geq T_{cloud}$ it is considered as 'cloudy',

otherwise (for $T_{clear} \leq RBR \leq T_{cloud}$), the pixel is said 'uncertain'.

For RAPACE imager, $T_{clear}$ =0.75 and $T_{cloud} = 0.85$. As an example, Long et al. (2006) used a unique threshold of 0.6.



- The differential tresholding process compares the RBR difference between the considered pixel and the corresponding pixel of a reference image, to 2 thresholds $Tdiff_{clear}$ and $Tdiff_{cloud}$:

  If $RBR - RBR_{lib} \leq Tdiff_{clear}$, the pixel is considered as 'blue',

  if $RBR - RBR_{lib} \geq Tdiff_{cloud}$ it is considered as 'cloudy',

  otherwise (for $Tdiff_{clear} \leq RBR - RBR_{lib} \leq Tdiff_{cloud}$), the pixel is said 'uncertain'.

  For RAPACE imager, $Tdiff_{clear} = 0.2$ and $Tdiff_{cloud} = 0.3$.

Figure 6 gives an example of an image processed with both the absolute (Fig. 6 c, e, g) and the differential (Fig. 6 d, f, h) processes. The results for this example are 78% (84%) of cloudy pixels, 15% (10%) of clear sky pixels, 7% (6%) of uncertain pixels for the absolute (resp. differential) process. The uncertain 'pixels' correspond to pixels that are difficult to define as clear sky or cloud. They usually correspond to thin cirrus, or to the border of a cloud. In this example, a stratocumulus cloud occupies most part of the image, but a thinner cirrus cloud above it can be seen in the lower part of the image. This thin cirrus is not defined as cloud by the absolute thresholding, but is partly identified through the uncertain pixels. However, it is entirely classified as a cloud by the differential process. Depending on the aim of an analysis, one may use one or the other result, or even utilise the difference for complementary information and detection of thin clouds.

**Figure 6.** (a) Cropped image of 13.02.2018 10:30 UTC. (b) Cropped reference image used for differential process, 12.02.2014 10:30 UTC. (c) RBR of cropped image (a) (for absolute process). (d) RBR difference between image (a) and reference image (b) (for differential process). (e) Result of Cloudy / Blue / uncertain pixels from absolute process. (f) Result of Cloudy / Blue / uncertain pixels from differential process. (g) pdf of the RBR in initial cropped image (see (c)). (h) pdf of the RBR difference between initial cropped image and referential cropped image (see (d)). In (g), the black dashed lines correspond to the thresholds $T_{clear} = 0.75$ and $T_{cloud} = 0.85$. In (h), they correspond to the thresholds $Tdiff_{clear} = 0.2$ and $Tdiff_{cloud} = 0.3$.





### 3.3 Adaptation to other cameras

This algorithm has been first developed for the RAPACE imager, which had no integrated process algorithm, and then adapted to other cameras of the ACTRIS-FR network of instrumented sites for homogeneity of the data process within the network. According to the systems, the image may differ in terms of geometry, color, obstructing objects,... Here are the camera-

dependent aspects that need to be adjusted or defined in ELIFAN for it to run on a new full sky camera:

- The geographical coordinates and type of system (fish-eye and camera): they determine the solar mask position and course along time;

- Specific additional masks, needed for certain cameras;

- The image center and radius, and the radius of the cropped image;

- the RBR pdf criteria which determine whether an image is fully clear, fully cloudy or partly cloudy (see step 3 above);

- RBR absolute and differential thresholding ratios, which vary from one camera to the other, and need to be optimized (see step 4 above).

The adaptation has been done for the EKO cameras listed in Tab. 1, and for the former TSI systems of SIRTA. The optimized thresholds used are indicated in Tab. 3. Those thresholds and those used in step 3 were optimized at $\pm\,0.05$ through a sensitivity

study based on one or two months of data for each camera. The raw images are automatically sent to the AERIS Icare Data Center at Lille, where ELIFAN runs and generates for each system, the corresponding libraries, the cloud cover netcdf data file, and the intermediate products (RBR, or RBR differences images like in Fig. 6 c and d, and the tricolour images of the cloudy/blue sky/uncertain pixels distribution like in Fig. 6 e and f).

**Table 3.** Thresholds used in step 4 for the different systems of the ACTRIS-FR network.

| Permanent site | Sky imager system | Cropped Image radius | $T_{clear}$ | $T_{cloud}$ | $Tdiff_{clear}$ | $Tdiff_{cloud}$ |
|---|---|---|---|---|---|---|
| P2OA | RAPACE | 545 | 0.75 | 0.85 | 0.2 | 0.30 |
| P2OA | EKO | 630 | 0.55 | 0.70 | 0.15 | 0.25 |
| SIRTA | TSI | 195 | 0.70 | 0.80 | 0.20 | 0.30 |
| SIRTA | EKO | 285 | 0.65 | 0.75 | 0.15 | 0.25 |
| CO-PDD | EKO | 300 | 0.65 | 0.75 | 0.15 | 0.25 |





### 3.4 Evaluation of the Algorithm, strengths and weaknesses

An entire year of RAPACE data (2014) has been used for ELIFAN direct evaluation, before its adaptation to other systems. Instead of considering all 15-min interval images, we considered only 1 image over 4 over the seasonal cycle, which resulted in a set of 4920 processed hourly images, which are thoroughly evaluated. The tricolour resulting images of the cloudy/blue

sky/uncertain spatial distribution of pixels were compared one by one to the initial images, and evaluated with human eye by a single operator. Several aspects were systematically considered for this evaluation, focused on specific previously identified potentiel failures :

- in detecting clear sky parts of the sky

- in detecting cloudy parts of the sky

- in detecting fully clear images (that is feeding the library)

- linked with thin cirrus clouds

- in the sunrise/sunset transition periods ($\pm$ 1 hour around sunrise or sunset time)

- due to the reflection and refraction of the sun

- due to rain drops on the Plexiglas dome

The main results of this evaluation are given in a synthetic way for the entire year in Tab. 4, through a percentage of successfull processed images over the relevant ensemble of images. The absolute thresholding process is considered first for several specific aspects. The differential thresholding process is evaluated with the relevance of all images classified as clear sky images and by checking that using the library improves the image analysis (and so the resulting cloud cover estimate).

**Table 4.** Eye evaluation of the performance of the algorithm: percentage of successfull processed images in various situations or aspects, over the entire year. The total number of images of the considered ensemble is specified in each specific evaluation.

| Evaluated aspect | Percentage of successfull images (%) | Total number of images in the considered ensemble | Considered ensemble |
|---|---|---|---|
| *Capability of detecting clouds or blue sky areas within the image (absolute thresholding, outside sunrise/sunset)* | | | |
| blue sky areas (%) | **99** | 2880 | All images with blue sky areas |
| cloudy sky areas (%) | **95** | 2836 | All images with cloudy area |
| *Capability in challenging situations (absolute thresholding)* | | | |
| cirrus clouds (%) | **64** | 1074 | All images with cirrus clouds |
| sunrise/sunset (%) | **13** | 639 | All images within sunrise/sunset period |
| *Relevance of the images within the library* | | | |
| clear sky image (%) | **77** | 554 | All images in clear sky library |
| *Improvement gained with differential thresholding* | | | |
| Improvement (%) | **54** | 338 | All images with a reference image |




From this human-eye evaluation, we deduced that out off dawn and twilight times, cloudy and clear parts of the sky are well identified by the absolute process, with respectively 95 and 99 % of success in the analysis (see Table 4). Condensation and rain drops have generally no significant impact on the cloud cover estimate, and the number of images that are significantly impacted is lower than 10 over the 4920 processed images. So ELIFAN works generally very well.

However, there are clearly identified weaknesses: Dawn and twilight periods (corresponding to 15% of the images) are critical times, with obvious failure of the algorithm, because "clear sky" is not "blue" then, and clouds are not "white or grey" (87% of failure, see Table 4). This is the main weakness of the algorithm, and main prospective of progress to put the efforts on. As a consequence, the blue sky reference library can also be improved: up to 23% of the images in the library are found inappropriate (see Table 4), a small half of them belonging to the critical dawn/twilight period (not shown).

We also find that 54% of the time, the differential process is judged better than the absolute process (see Table 4). From this results, one may conclude that there is no obvious advantage of the differential process relative to the absolute process. But it definitely gives better results in detecting clear sky areas in the circumsolar region and in detecting thin cirrus clouds (which will be illustrated later). Furthermore, the use of the combination of both absolute and differential processes may interestingly be used for the estimate of thin cirrus clouds.

In addition to the previous systematic evaluation, a general statistical analysis over one year of data without decimating (more than 13000 images) was made on the departure between the cloud fraction deduced from the absolute thresholding process ($CF_{abs}$) and the cloud fraction deduced from the differential thresholding process ($CF_{diff}$). The median of the difference in percentage of cloud is 2%. The 10% and 90% quantiles of the departure are respectively at -7% and 9%. 8% of the values of $|CF_{diff} - CF_{abs}|$ are larger than 10%. And this latter number drops to 3% of the images only when we consider only the period of the day between 0900 UTC and 1500 UTC, that is when avoiding the tricky period of sunrise and sunset. Thus, large values of the departure $|CF_{diff} - CF_{abs}|$ are associated to this transition period issue, to thin cirrus occurrence, and to other specific situations that would lead to outliers. Figure 7 shows the correlation between $CF_{diff}$ and $CF_{abs}$ for this same large set of images. Considering only the midday period (and so avoiding the dawn/twilight period, see blue color in Fig. 7) improves the correlation and 1-to-1 correspondence, and especially reduces the under-estimations by $CF_{diff}$ relatively to $CF_{abs}$.





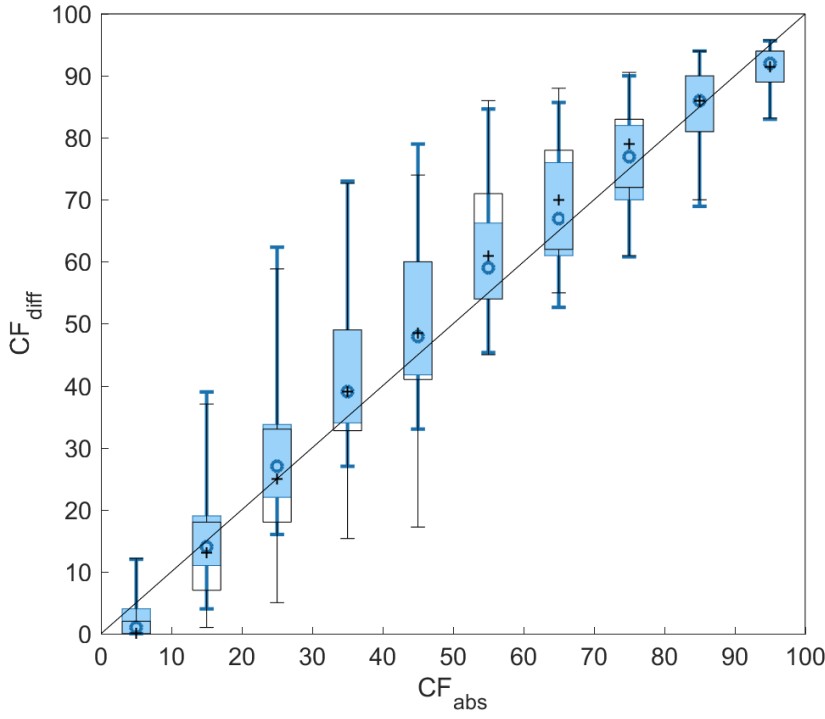

**Figure 7.** Comparison the cloud fraction estimates deduced from the two thresholding processes, over a set of more than 13000 images taken during one year, (black) at any time of the day and (blue) only during midday (between 0900 and 1500 UTC). $CF_{diff}$ as a function of $CF_{abs}$ is shown through boxplots changing with bins of values of $CF_{abs}$, with 10% increments. boxes represent the [25-75] percentiles interval, and the vertical lines show the limits of the 5 and 95 percentiles. The symbol into the box stands for the median.



## 4  Illustrating examples with the analysis of cloud cover series

Here we illustrate the gain obtained by the observation of a sky imager at an observatory, complementarily to ceilometer and pyranometer typical measurements, as well as the performances of ELIFAN.

Two days are considered here, the 5 June 2016 (in Fig. 8) and the 9 June 2016 (in Fig. 9), where we compare the cloud cover estimates from ELIFAN, and the cloud cover estimates from a CT-25K ceilometer, at P2OA site. The latter have been calculated over 15, 30 and 60 min intervals, based on the occurrence of detection of a cloud base by the ceilometer during this time interval. Complementarily, we also use CNR-4 Kipp&Zonen pyranometer measurements, in order to get some information on the downward shortwave radiation. In Figs. 8 and 9 we show, for three raw RAPACE images at chosen times (panels (a), (b) and (c)), the ceilometer cloud base height detection (panel (d)), the cloud cover estimates (panel (e)), and the shortwave downward radiation (panel (f)). Note that we do not expect the cloud cover estimates deduced from the two instruments to entirely agree, since they are based on very different concept. As overlined by Wagner and Kleissl (2016) before, the ceilometer sees a limited area of the sky above, relatively to the sky camera which integrates a larger 2D area of sky above, and a limited range of measurement along the vertical (maximum 7-8 km a. g. l.) depending on solar/aerosol conditions. We nevertheless expect them to compare consistently together. Complementarily, the ceilometer brings additional information on the cloud base height, and the pyanometer gives information on the radiative impact of the clouds at surface.

On 5 June 2016, the day started with a full deck of low stratus clouds, which progressively broke and lifted during the morning convection, until dissipating totally later in the afternoon (see Figs. 8, all panels). This is a typical situation when the algorithm works very well, and also when we find a large consistency between the ceilometer-based cloud fraction and the sky imager-based cloud fraction, as seen in Figs. 8(e). Looking at carefully some chosen times of the day, one can tell more about each phase.

At 07:00 UTC, the cloud cover is 100% (see Figs. 8(a)). The ceilometer-based fraction and the absolute ELIFAN thresholding cloud fraction agree on this. However, the ELIFAN differential thresholding cloud fraction is slightly below 100%, around 95%. This could seem paradoxical, but is due to the large luminosity to the east (right in the image Figs. 8(a)), that is close to the white color found in circumsolar area of the reference clear sky image (even out off the already important solar mask). So the 100% cloud cover are indeed better caught by the absolute thresholding method. (That is, the expected improvement around the sun is actually obtained in case of clear sky, but not in case of cloudy sky like here.)

Starting 15:45, the sky is almost clear (only small cumulus in Figs. 8(c)), the cloud cover should be very close to 0%. The ceilometer-based fraction and the differential ELIFAN thresholding cloud fraction agree on this fully clear sky. The absolute thresholding cloud fraction leads to a non-zero amount during this period, below 5%. This is due to both the small clouds north to the instruments and not caught by the ceilometer that indeed make the cloud fraction non-zero, but in a larger extent to the circumsolar area, around the solar mask, that appears as a cloud.

After 18:00 UTC, a large discrepancy is found between the ceilometer and the sky imager estimates. This is due to the transition period issue mentioned in section 3. The absolute thresholding technique especially fails in seeing the clear sky. The differential thresholding is doing a little better. The ceilometer, which is a priori not affected by sunset or sunrise, does not see



any cloud. The original RAPACE images and the downward SW (Figs. 8(f)) actually show some thin remaining and spreading clouds to the north (close to the mountain ridge). But ELIFAN differential estimates still probably over-estimate the cover. One can see here that whatever the instrument (and even with eye), this transition period remains delicate to analyse, and may remain quite uncertain.

The example shown at 11:45 UTC (Figs. 8(b)) is the typical situation of very good and consistent results of both the absolute and the ELIFAN differential thresholding techniques. The sensitivity study on the time intervals used to calculate the ceilometer-based cloud fraction estimates show that the 30 min interval seems appropriate and consistent with the sky imager for that case. Between 8:00 and 11:00, the cloud field is similar to Figs. 8(b) (low cumulus clouds), but with sometimes some spatial heterogeneity (not shown). The estimates from the ceilometer can vary a lot, and can also differ significantly from

the ELIFAN estimate. This is due to this spatial heterogeneity, that the ELIFAN estimates will integrate in its 2D approach while the ceilometer is restricted to a 1D approach along wind. Those results are fully consistent with Wagner and Kleissl (2016), who have evaluated the ceilometer cloud cover retrievals by comparison with sky imager retrievals, and pointed out the heterogeneity of the cloud field as one of the sources of errors in ceilometer-based estimates.

The second example shows more obviously the different information that each instrument brings, and how complementary

they are.





(a)    (b)    (c)

(d)

(e)

(f)

**Figure 8.** First row : Three RAPACE images on 5 June 2016 at (a) 08:00 UTC, (b) 11:45 UTC, (c) 17:00 UTC. (d) Cloud base heights measured by the ceilometer, (e) Cloud cover estimates from the ceilometer and from RAPACE sky imager, and (f) Downward shortwave radiation, on the same day. In (e), several integrating time intervals are considered for the ceilometer-based estimates: (black) 15 min, (blue) 30 min, (cyan) 1 h. Results from both the (red) absolute thresholding and (magenta) differential thresholding are shown for the sky-imager-based estimates. In (f), both the (gray) 10 s initial pyranometer measurements and the (dark gray) 15-min averaged data are shown. The background blue sky observed on a clear day during the same month is shown in cyan color. In (d), (e) and (f), the time when the pictures (a), (b) and (c) were taken is indicated by an orange vertical line.





**Figure 9.** First row : Three RAPACE images on 9 June 2016 at (a) 07:00 UTC, (b) 11:00 UTC, (c) 15:00 UTC. (d) Cloud base heights measured by the ceilometer, (e) Cloud cover estimates from the ceilometer and from RAPACE sky imager, and (f) Downward shortwave radiation, on the same day. In (e), several integrating time intervals are considered for the ceilometer-based estimates: (black) 15 min, (blue) 30 min, (cyan) 1 h. Results from both the (red) absolute thresholding and (magenta) differential thresholding are shown for the sky-imager-based estimates. In (f), both the (gray) 10 s initial pyranometer measurements and the (dark gray) 15-min averaged data are shown. The background blue sky observed on a clear day during the same month is shown in cyan color. In (d), (e) and (f), the time when the pictures (a), (b) and (c) were taken is indicated by an orange vertical line.



9 June 2016 has a complex sky, with a mix of mid-level clouds, and -thinner or deeper- cirrus clouds (Fig. 9(a), (b), (c), (d)). This is a typical case of potential difficulties in cloud cover estimates, and even potentially on cloud cover amount definition. The sky imager and ceilometer cloud fraction estimates differ a lot during that day (Fig. 9(e)).

Around 07:00 UTC, the thin cirrus clouds shown by RAPACE sky camera are not detected at all by the ceilometer, while they are visible in several areas (Fig. 9(a)) and have a significative impact on the shortwave downward radiation (Fig. 9(f)). It could be explained by the fact that the cirrus clouds are too high (above 7 km) for the ceilometer sensitivity. The cloud cover amount found by ELIFAN from the sky camera images (around 40%) makes sense, and is more appropriate than the ceilometer 0% cloud cover estimate. This is also fully consistent with the work done by Wagner and Kleissl (2016) who identified the limitation of the vertical range as the largest source of error on the ceilometer-based estimates (more significant than the error linked with the heterogeneity in the cloud field, mentioned in the previous example).

During this morning period of 9 June 2016, the differential and the absolute thresholding estimates are sometimes significantly different: this is due to the presence of very thin cirrus clouds, that are not caught by the absolute thresholding. Thus, the difference between both estimates gives an indirect estimate of the amount of very thin cirrus clouds (it is also the case at 16:30 UTC for example). The impact of very thin cirrus clouds on the downward shortwave radiation is not negligible, about 100 $\mathrm{Wm}^{-2}$ (Fig. 9(f)), which means that the differential thresholding is more relevant than the absolute thresholding in case of thin cirrus.

At 11:00 UTC, the cirrus got larger and also slightly deeper, and some mid-clouds start to appear as well, all this leading to a full covering of the sky. This explains why both the ELIFAN differential and the absolute thresholding techniques agree with a large cloud cover (close to 100%). The impact of those clouds on the downward shortwave radiation is more significant than at 07:00, with about 40% reduction of solar energy (Fig. 9(f)). The ceilometer detects the cirrus more than before as well as the mid-level clouds. But the ceilometer cloud fraction estimates is 20 %, which still seems underestimated.

At 15:00 UTC, the mid-level clouds cover more largely the sky, screening the detection of the cirrus clouds by the ceilometer. The ceilometer-based cloud fraction is consequently increased, larger than 60%, but still lower than the sky imager estimates. The impact of the cloud cover is significant on the downward shortwave radiation.

The dawn and twilight transitions are again problematic here for ELIFAN : starting at 19:00 UTC, the thickness of the cirrus clouds gets still larger then before, without mid-level clouds below. The ceilometer-based estimates increase, but ELIFAN interpretes the images as clear sky. Around 05:00 UTC, the sky is clear (see Fig. 9(f)), the ceilometer and the ELIFAN differential thresholding lead to 0% cloud cover, but the absolute thresholding techniques fails and finds non-zero cloud cover (up to 20%).

## 5   Concluding remarks

From the evaluation of ELIFAN on a one-year long data evaluation, we deduced that outside the morning sunrise and sunset transitions, ELIFAN is a robust and efficient algorithm to evaluate the cloud cover amount from a full sky camera. 95% of the daytime cloudy images and 99% of clear sky images are analysed appropriately. So ELIFAN works generally very well. It is also relatively easy to adapt to a new camera, as soon as the quality of the image is high enough.



However, the sunrise/sunset transition is the main weakness of the algorithm. This is where efforts should be done in the future, for improvement of ELIFAN. Specific criteria and thresholding could be applied in this time window.

The effort of developping a common sky-image data process for a network of sky imager has been very fruitful, as now 5 cameras of the ACTRIS-FR French multi-instrumented sites network have their images processed and gathered together at the

5 same nation Data Center of AERIS-Icare (at https://actris.aeris-data.fr/data/). Three more cameras (on three other sites) should join soon. It is planned to unvalidate the sunrise/sunset period and the erroneous images of the library, before opening free access to consolidated data, and until improvement of the algorithm during this specific transition period.

The use of the clear sky library in the differential thresholding process generally improves the image process relatively to the absolute process, since it better detects thin cirrus clouds. But the absolute thresholding process turns out to be sometimes

better, for example in fully cloudy skies. It is also definitely useful for days with no reference image, or in case of casual field experiments where there is no clear sky library available. Moreover, the combination of both estimates may be an interesting way to access to the estimate of the thin cirrus clouds. But this remains to be further evaluated.

The combination of ceilometer and sky camera has revealed the complementarity of both instruments. We showed that a similar cloud cover amount is deduced from both instruments when the clouds are low. But they lead to very different cloud

cover estimates when the clouds are too high for the ceilometer vertical detectability range, or when the field of cloud is heterogeneous. Those results are in full agreement with the recent work made by Wagner and Kleissl (2016).

Indeed, the two instruments have fundamentally different points of view, as the ceilometer points vertically and the integration time considered for the cloud cover estimate only samples a small part of the sky above. While the sky camera has a full 2D view. Thus, the sky imager catches more appropriately the spatial variability, and the cloud cover estimates from its images

are more representative of the real sky. Also, the sky camera can localize the clouds in the sky, which can be very useful in solar energy nowcasting.

The clear advantage of the ceilometer though, lies in its capability of quantifying precisely the cloud base height(s). So that the two instruments are very complementary. The information brought by the ensemble of a ceilometer, radiometers and a full sky camera together is shown to be a very relevant synergy of instruments.

*Acknowledgements.* The data of RAPACE sky imager, the ceilometer and the pyranometer used here were collected at the Pyrenean Platform for the Observation of the Atmosphere P2OA (http://p2oa.aero.obs-mip.fr). P2OA facilities and staff are funded and supported by the University Paul Sabatier, Toulouse, France, and CNRS (Centre National de la Recherche Scientifique). We thank Météo-France for the ceilometer installation at P2OA and for the data availability. We thank AERIS data infrastructure for development support and data processing, and especially the ICARE Data and Services Center. ACTRIS-FR and AERIS are supported by the French Ministry of Education and

Research, CNRS, Météo-France, CEA (Centre d'Energie Atomique), INERIS (Institut national de l'environnement industriel et des risques). The developement of ELIFAN algorithm has been funded by the University Paul Sabatier of Toulouse - Observatoire Midi-Pyrénées, and by ACTRIS-FR. We thank Franck Gabarot and Alain Sarkissian for information and discusions about the sky cameras at OPAR and OHP instrumented sites.





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
