# Peer review of "ELIFAN, an algorithm for the estimation of cloud cover from sky imagers"

_Atmospheric Measurement Techniques, 2018_

## Referee Comment (RC1) · Anonymous Referee #2 · 5 Mar 2019

General comments:

This manuscript presents an algorithm named ELIFAN for the estimation of cloud fraction from sky imagers. The evaluation based on one-year of RAPACE data with human eyes shows the good performance in cloud identification at the P2OA plain site. The results and conclusions are presented clearly and concisely. Cloud identification is the basis of estimation of cloud fraction. This paper has discussed the main weakness of ELIFAN during sunrise/sunset transition and the shortage in thin cirrus clouds. However, besides the thin cirrus clouds, aerosols and fogs are two big problems challenging the accuracy of cloud identification for visible images. Discussions about impacts of aerosols and fogs on ELIFAN are not presented in this paper. Since there are some obvious weaknesses in the innovation, methodology, and validation of ELIFAN, major

revisions are suggested.

Specific comments:

1) This study focuses on the algorithm. Since there have been some algorithms of estimating cloud cover from all sky images, progress or advances of ELIFAN relative to previous researches or other algorithms should be addressed clearly in the introduction or section 3.1.

2) The ELIFAN contains an absolute and a differential threshold process to estimate the cloud cover. Is ELIFAN automatic? When the two cloud covers are different, which one will be used in "real-time" operation? More words should be added.

3) For the differential threshold method, how does the reference image be selected for an all sky image? How do you deal with the differences caused by solar position and background atmosphere which change with the time and location? In addition, settings of white-balance mode also exert influences on colors of images. Please add more statements of these problems.

4) There exist some thin clouds (near the solar circle, area around (400,700)) in Figure 5(a). Why do the authors think it is a clear sky?

5) Aerosols and fogs always show similar R/B features with clouds in visible images. They challenge the accuracy of cloud identification, especially for skies of low visibility. How about the visibility (or aerosol optical depth) of those days for the validation in section 3.4? How about the occurrence frequency of thin cirrus clouds? The performances of ELIFAN are dependent on the sky conditions of images. Please present some rough expressions of the sky conditions of days for validation.

6) What is the purpose of section 4? If it is aimed to show the performance of ELIFAN via comparisons with the pyranometer and ceilometer, the results are destined to be weak since the two instruments work in a different way. The pyranometer cannot estimate cloud cover. The ceilometer can estimate the occurrence frequency of

clouds during a period. Supposing that the formation of clouds is random, the occurrence frequency during a certain period might be regarded as cloud cover. However, the formations, evolutions and movements of clouds interact with the atmospheric and topographical conditions and show regional character. The differences between occurrence frequency and cloud cover change due to different atmospheric conditions and locations. Thus, it's deficient to deduce the strength or weakness of ELIFAN through the comparison. If it is aimed to show the complementarity of all sky camera and the ceilometer, the work somewhat departs from what the title indicates. Maybe, add "and its application" in the title to keep this section.

7) How about the transferability of ELIFAN? Is it applicable for other areas, for example areas of high aerosol optical depth? Do you have some special approaches to discriminate the aerosols or fogs from clouds? More discussions should be added.

---

## Referee Comment (RC2) · Anonymous Referee #1 · 12 May 2019

The manuscript deals with the development of an algorithm for the estimation of cloud coverage from a network of different all-sky imagers. Although the estimation of cloud coverage is the basic metric derived from an all-sky imagers (in literature there are a lot of methods for estimating several optical and geometrical properties of atmospheric constituents), the need to investigate it is valid for 2 reasons: first, it may provide much better estimations than visual observations; second, a common algorithm is needed in order to build a homogenized network of instruments with comparable results.

However, the physical base to characterize an image pixel as cloudy or not, is the radiometric calibration of the all-sky imager. This is not the case here. Instead, the well-known solution of building a clear-sky library is proposed. So, we have to be precise here: we are not talking about clear sky images but for cloud-free ones. Aerosols

are not considered and the same stands for thin cirrus clouds. As a result, it is sure that thin cloudiness is not estimated correctly. However, due to the fact that there is a real scientific need to develop and test such algorithms, this original error could be neglected. Even under this consideration, the manuscript needs major changes before published, as follows:

Most of the questions arise from the description of the methodology steps (paragraph 3.2.1).

1. How the circumsolar area is selected? Is it always the same area around the Sun? Is it dependent on the solar zenith angle (the Sun disk area, depicted in the image is dependent for sure).

2. The cropped area does not refer only to obstacles (e.g. the building and the surroundings cover only a very small part of this area) but also to droplet area and clouds at high solar zenith angles. This may improve the results a lot because this is the "difficult area" for cloud cover decision. What is the percentage (relative to the total sky are) of the cropped area?

3. Please add more details in step 2 (page 9, lines 3-5). In order to put the solar mask on the image around the position of the Sun, you need some kind of geometric calibration.

4. Page 9, line 16-17: it is mentioned that "if there are several reference images, the sky with the least turbidity is chosen as reference, based on the RBR pdf". If the other references images are not used, they have to be moved from the clear sky library before the classification. The use of the image with the minimum turbidity cannot be considered as the most important one as it does not correspond to the "average" turbidity (or aerosol) conditions. It is exceptional for sure. So, another approach (e.g. an image selection for the "average" conditions) should be followed.

Some more comments for the rest of the manuscript:

5. Figure 5a: this is definitely not a clear sky image. It may classified as clear sky but it is not and this should be discussed. Moreover, it is concluded later that the differential method is judged better than the absolute and definitely gives better results in detecting thin cirrus clouds (page 17, lines 10-14). So, why you do not use the differential method instead of the absolute also here for the categorization of such a type of images?

6. Page 12, line 20: it is not clear if the method is dependent for solar zenith angle. The images in figure 3 correspond almost to the same solar zenith angle. What about a set of images at a lower solar zenith angle? Are the same thresholds valid? Moreover, it is very common that the RBR values change among different cameras. Apart from the distribution of RBR, which is the effect on RBR threshold values?

7. Page 12, lines 22-23: Why do you choose now the pixel by pixel process but not in the previous step? In this way, some cloudy areas (e.g. in figure 5a) could be removed from the "cloud-free" image. 8. Figure 6 and relevant text: The sky image in figure 6b has been detected as partly cloudy.

However, it is almost identical with the image 5a. Which is the pdf versus RBR for the 6b image? This example questions strongly the robustness of the cloud-free image selection method.

9. Table 3and relevant text: the EKO camera at P2OA presents different Tclear and Tcloudy values when compared to those at SIRTA and CO-PDD. Please explain the differences taking into account that the cropped radii are different.

10. Page 17, line 20: the hours between 9 and 15UT corresponds to which solar zenith angles in summer and winter time? How this is related to the cropped sky area?

11. Section 4: It is good to know that an all-sky imager could provide useful information for cloudiness at any measuring site. It also explains much of the measured variability of surface irradiance. This seems to be a nice paragraph promoting all-sky imagers but it does not enhance the validity of the algorithm. Moreover, the fact that an all-sky

imager, compared to a ceilometer, could provide a holistic information for cloud cover (but not for cloud height) is totally expected. It is not clear what is the purpose of section 4, it may be removed.

12. Based on the above comments, the abstract and conclusion paragraphs should be revised.

Minor comments: Page 9, line 4: the solar mask (instead of maks)

Figure 3: please align horizontally the 2 images

Figure 5: please check again the y axis title in figures d,e,f.

---

## Author Comment (AC1) · 2 Jul 2019

**Referee #2**

We deeply thank the reviewer for his/her comments and questions, which helped to improve the clarity of the manuscript.
Below are the responses to each of his/her comments.

But first, note that the values of the evaluation in Table 4 have changed. We have found that at submission, we had not taken into account the most updated and optimized evaluation that had been previously done. This does not change any of the concluding messages in the article.

**General comments:**

However, besides the thin cirrus clouds, aerosols and fogs are two big problems challenging the accuracy of cloud identification for visible images. Discussions about impacts
of aerosols and fogs on ELIFAN are not presented in this paper.
→ **Yes, this is true. We do not consider the influence of aerosol at all in this algorithm.**
**It turns out that the algorithm performs quite well without doing so.**
**But it could be different in region with much more aerosol loading.**
**Fog is not addressed here either. But ELIFAN is detecting 100 % cloud cover during the fog events, which remains a correct cloud fraction estimate in this case.**
**This was tested with images of fog events, and also quantified at SIRTA with a visibilimeter :**
**All fog cases studied (visibility < 1000 m) were found with 100 % cloud cover with ELIFAN.**

Since there are some
obvious weaknesses in the innovation, methodology, and validation of ELIFAN, major revisions are suggested.

**Specific comments:**

1) This study focuses on the algorithm. Since there have been some algorithms of estimating cloud cover from all sky images, progress or advances of ELIFAN relative to previous researches or other algorithms should be addressed clearly in the introduction
or section 3.1
→ **This was actually partly done in the introduction of section 3.2, but only after the description of different methodologies found in the litterature. We added on page 8, lines 10-12 some more comments on what ELIFAN brings to those previous works, even if there is no fundamental innovation relatively to the principle of the method used here :**

**One originality of ELIFAN, is that it applies both an absolute and a differential thresholding processes independently. Each of them has advantages and drawbacks, but both are complementary.**

2) The ELIFAN contains an absolute and a differential threshold process to estimate
the cloud cover.
Is ELIFAN automatic?
→ **Yes.**

When the two cloud covers are different, which one will be used in "real-time" operation?

More words should be added.

**→ Both are estimated and available. The user is free to use one or the other estimate, depending on the goal of his/her study, and taking account of the strength and weaknesses of each process.**
**See page 12, lines 13-14 of the manucript « Depending on the aim of an analysis, one may use one or the other result, ».**

3) For the differential threshold method, how does the reference image be selected for an all sky image?
How do you deal with the differences caused by solar position and background atmosphere which change with the time and location?
**→ The reference images are selected as cloud free images with the same solar zenith and azimuth angles than the processed image. This was indicated in the initial manuscript.**
**See page 8, lines 20-21 of the revised manucript : « the algorithm searches for a reference blue sky image within a library, with the sun at same azimut and same elevation, ±1° , as the considered image » [i.e. as the image to be processed].**

**If there are several reference images with the same zenith and azimuth angles, the image with the least turbidity is used (that is the one with the PDF having the smallest RBR). This was also indicated in the initial manuscript.**
**See page 8, lines 24-25 of the revised manucript : « If there are several reference images available, the sky with the least turbidity is chosen as a reference, based on the RBR pdf. »**
**But, following Rev#1 suggestion, we have changed the algorithm to rather use the image with the median turbidity (median RBR distribution), which is indeed more appropriate. This will be part of the next version of ELIFAN.**
**We added a discussion about this in the conclusion.**

In addition, settings of white-balance mode also exert influences on colors of images.
Please add more statements of these problems.
**→ We did not consider this aspect at all. As far as we know, we did not need to adjust the white-balance of the images before adapting ELIFAN to the various cameras. But this is a relevant suggestion, especially for cameras which take images with strong differences in this setting.**

4) There exist some thin clouds (near the solar circle, area around (400,700)) in Figure 5(a). Why do the authors think it is a clear sky?
**→ We agree. Image 5a is not absolutely cloud-free. As an example of a cloud free image, with the corresponding pdf, we do need to give a better illustrating example.**
**We have now considered another time with a fully cloud-free image, as a better example.**
**We also corrected an error on the year of the images, which was 2014 (not 2018).**

5) Aerosols and fogs always show similar R/B features with clouds in visible images.
They challenge the accuracy of cloud identification, especially for skies of low visibility.
How about the visibility (or aerosol optical depth) of those days for the validation in section 3.4?
**→ We did not consider fog or aerosol neither in the algorithm, nor in our evaluation.**
**Note that fog and aerosol profiles are not measured on all sites. It is on sites like SIRTA, and could be an aspect to address in the future with this site.**
**For now, fog cases, and low visibility of very lowcloud decks are estimated as 100 % cloudy images by ELIFAN.**

**But we do not depart low clouds from fog, and to not estimate any index of visibility with ELIFAN.**
**More over, ELIFAN does not aim at identifying the type of clouds.**

How about the occurrence frequency of thin cirrus clouds?
**Thin Ci clouds occurrence has not been estimated during the human eye evaluation.**
**But Table 4 shows that we have 1356 images with some cirrus clouds, among the 4925 considered images. The success rate in identifying those Ci is 78%. This means that potentially 22% of those 1356 images are thin Ci clouds, that is 6 % of the 4925 images. Assuming that some of the thin cirrus are well caught, 6 % is a minimum estimate of thin Ci occurrence.**

The performances of ELIFAN are dependent on the sky conditions of images. Please present some rough expressions of the sky conditions of days for validation.
**→ It is not easy to answer this question, because the 4925 images were picked up all along the year with a systematic and fixed time interval.**
**That means that most of the conditions which can be encountered in the (middle-latitude) site are considered.**

6) What is the purpose of section 4?
**→ This question was also raised by Rev #1.**
**The main goal was to illustrate the algorithm process presented before, and discuss it with other points of view brought by complementary instruments.**
**Which eventually gives 2 objectives :**
**(1) illustrating the ELIFAN process and weakness/strength points presented before**
**(2) showing the complementarity of the instruments.**
**We thought that this section would be useful, especially for readers who are interested in the application of the sky cameras for process studies.**

**But, based on the fact that :**
**- both Reviewer #1 and Reviwer #2 have put this section into question,**
**- we have inserted more illustrations of the strength and weakness of ELIFAN in section 3,**
**- it is appropriate to keep the manuscript to a reasonable length,**
**we propose to remove this section.**
**With this change, the previous evaluation section 3.4 has now been renamed section 4 itself, in order to better balance the size of the various parts of the revised article.**

**However, we are still willing to keep the original section 4 if ever the Reviewers and Editor finally find it more appropriate.**

If it is aimed to show the performance of ELIFAN via comparisons with the pyranometer and ceilometer, the results are destined to be weak since the two instruments work in a different way.
**→ We fully agree with this.**

The pyranometer cannot estimate cloud cover. The ceilometer can estimate the occurrence frequency of clouds during a period. Supposing that the formation of clouds is random, the occurrence frequency during a certain period might be regarded as cloud cover. However, the formations, evolutions and movements of clouds interact with the atmospheric and topographical conditions and show regional character. The differences between occurrence frequency and cloud cover change due to different atmospheric conditions and locations. Thus, it's deficient to deduce the strength or weakness of ELIFAN through

the comparison.
**We agree with this too. That was not our purpose.**

If it is aimed to show the complementarity of all sky camera and the
ceilometer, the work somewhat departs from what the title indicates. Maybe, add "and
its application" in the title to keep this section.
**→ Yes. If we were to keep this section, we would adapt the title, and more clearly explain the aim of the section in the start.**

7) How about the transferability of ELIFAN? Is it applicable for other areas, for exam-
ple areas of high aerosol optical depth?
**ELIFAN is transferable to other areas, but should be difficult to use in regions with heavy aerosol loading, since it does not deal with this aspect.**
**Also it is potentially more challenging to use it in Tropical regions, because the sun is for a large part of the time in the center of the image, which is the most « useful »/ »easy » part of the image for the process.**
**We added some comments in the conclusion about this.**

Do you have some special approaches to discriminate the aerosols or fogs from clouds?
More discussions should be added.
**→ Using dynamical thresholds may do a better job in sunrise/sunset transitions, and also for aerosol impact on the variability of the cloud-free sky.**
**We do not plan to attempt to depart fog or aerosol from clouds with ELIFAN. We believe that ceilometer, lidar, and visibilimeters should do better, and are really complementary of all-sky imagers.**
**The perspective of using dynamical thresholds is mentioned in the conclusion.**

---

## Author Comment (AC2) · 2 Jul 2019

**Referee #1**

We deeply thank the reviewer for his/her comments and questions, which helped to improve the clarity of the manuscript.
Below are the responses to each of his/her comments.

But first, note that the values of the evaluation in Table 4 have changed. We have found that at submission, we had not taken into account the most updated and optimized evaluation that had been previously done. This does not change any of the concluding messages in the article.

[…] we have to be precise here: we are not talking about **clear sky** images but for **cloud-free** ones. Aerosols are not considered and the same stands for thin cirrus clouds.
→ **Yes, we understand. Although « clear sky » is quite generally used in the litterature, we agree that we rigorously should use « cloud free ». We made the changes accordingly.**

1. How the circumsolar area is selected?

→ **The sun mask diameter in pixels is a compromise between loosing some processed pixels (when we enlarge the sun mask), and increasing the difficulty/error (when we make itsmaller, and so add impacted pixels).**
**We have specified this in the revised manuscript, in the description of step 2 page 8, lines 10-12.**

Is it always the same area around the Sun?
→ **yes. But when the sun is at the border (higher zenith angle), part of the mask gets outside the processed image. So it ends with a different portion of the total processed image.**

2. The cropped area does not refer only to obstacles (e.g. the building and the sur-
roundings cover only a very small part of this area)
but also to droplet area and clouds at high solar zenith angles.
This may improve the results a lot because this is the "difficult area" for cloud cover decision.

What is the percentage (relative to the total sky are) of the cropped area?
→ **The cropped area is fixed, and corresponds to an aperture angle of 143° centered toward zenith.**
**It avoids obstacles close to the image border, and also the sky close to horizon which can not be properly interpreted : a 4/8 cumulus field would be erroneously found 8/8 at horizon when counting the cloudy pixels of the image, because of this low elevation angle of view. It does not deal with cloud droplets.**
**We have specified the aperture angle of the cropped area in the revised version, page 7 line16.**

3. Please add more details in step 2 (page 9, lines 3-5). In order to put the solar
mask on the image around the position of the Sun, you need some kind of geometric
calibration.

→ **The solar zenith and azimuth angles are calculated from the method given by Reda, I., Andreas, A. (2003) : « Solar position algorithm for solar radiation application. National Renewable Energy Laboratory (NREL), Technical report NREL/TP-560-34302 ».**

**If we write $I_s$, $J_s$ the coordinates of the sun in the image (units = pixels) :**

*$I_s$=Int (A − B sin (α/2 ) cos β )*
*$J_s$=Int (A − B sin (α/2 ) sin β )*

*Where A and B are adjusted according to the camera (A is close to the diameter of the cropped image).*

*This information has been added in the description of step2, page 8.*

**The accuracy of this position is quite low though, of about 5°. Due to the large sun mask, it does not significantly impact on the results.**
**Nevertheless, we have now considered a more accurate positionning, based on Crispel and Roberts, 2018, and the Lambert projection for the representation function that takes account of the fish eye deformation. This leads to a calibrated fourth order polynomial for Is and Js. This will be implemented in the next version of ELIFAN. We added this information in the conclusion.**

4. Page 9, line 16-17: it is mentioned that "if there are several reference images,
the sky with the least turbidity is chosen as reference, based on the RBR pdf".
If the other references images are not used, they have to be moved from the cloud free
library before the classification.
The use of the image with the minimum turbidity cannot be considered as the most important one as it does not correspond to the "average" turbidity (or aerosol) conditions. It is exceptional for sure.
So, another approach (e.g. an image selection for the "average" conditions) should be followed.

**→ Yes. It is true that using the image with the least turbidity may not be the most appropriate. We have now considered the possibility of using the image with the median amount of turbidity, among the set of reference images, for the next version of the algorithm. We mentioned this in the conclusion of the revised version.**

**As an illustration, we show below a set of reference of cloud-free images with same zenith and azimuth angles. We can see here that there is not a very large spectrum of variability in this case of RAPACE camera images.**

[Figure]

|  |  |  |  |  |  |
|---|---|---|---|---|---|
| 2013/12/15 1330 UTC | 2014/12/22 1330 UTC | 2015/12/19 1330 UTC | 2015/12/22 1330 UTC | 2017/12/25 1330 UTC | 2018/12/19 1330 UTC |

[Figure]

*PDF of the Red over Blue ratio for a set of reference images of same solar zenith and azimuth angle (see icones above).*

**Below is an example of the result of the differential threshold process in the case of using the image with the least turbidity, versus using that of median turbidiy, for an early June case. For the considered image, there are thin cirrus in the middle of the image, which can be more sensitive than other clouds to the library image used. But the difference between the two processes is small. The main difference is found around the sun mask. The difference in CF is of 4% in this example.**

[Figure]

| Cropped image
2016/06/03
0830 UTC | Differential process based on minimum turbidity reference image | Differential process based on median turbidity reference image |

5. Figure 5a: this is definitely not a cloud free image. It may be classified as cloud free but it is not and this should be discussed.
 **→ We agree that Image 5a should be changed. As an example of a cloud free image, with the corresponding pdf, we do need to give a better illustrating example.**

**We have now considered another time with a fully cloud-free image, as a better example. We also corrected an error on the year of the images, which was 2014 (not 2018).**

Moreover, it is concluded later that the differential
method is judged better than the absolute and definitely gives better results in detecting
thin cirrus clouds (page 17, lines 10-14).
**→ Yes, this was the result of a close look of the many images for evaluation, and sorted as a very general result. But the answer really depends on the conditions/images.**

**Below is an example (same as in question 4 above) with some thin cirrus clouds, which are quite extended in the middle of the image. Deeper Ci clouds surround them. The absolute threshold finds a large number of uncertain pixels within the thin cirrus clouds (see the area around X=400 and Y=600 in the middle panel figure below), while the absolute threshold process will consider them as clouds.**
**So if one wants to consider those clouds into the overall cloud cover, the differential process is doing relatively better here. While the amount of uncertain pixels in the absolute process gives an estimate of the thin cirrus cover.**
**Finally, one can also see in this example that in the circumsolar area, the absolute threshold gives better result in this area.**

[Figure]

Cropped image         Absolute threshold process      Differential threshold process
2016/06/03
0830 UTC

**Below are 2 other examples of the impact of the circumsolar region.**
**One where the sun shines through a cloud-free area, and one where it is is hidden by clouds.**

**In the first case, the use of the reference images makes a better interpretation of the circumsolar region. But in the second case, the effect is reverse, and the absolute process works better in this region. It also works generally better in detecting the cloud field of this image.**

[Figure]

[Figure]

| Cropped image | Absolute threshold process | Differential process |
| --- | --- | --- |
| 2018/02/10 | | |
| 1300 UTC | | |

[Figure]

[Figure]

| Cropped image | Absolute threshold process | Differential process |
| --- | --- | --- |
| 2018/03/07 | | |
| 1100 UTC | | |

**We have considered some of those examples in the revised version, in order to better illustrate the discussion on strength/weakness of ELIFAN, and the different results of both processes (see page 16 , lines 17-28).**

So, why you do not use the differential method instead of the absolute also here for the categorization of such a type of images?
**→ The library images are by construction not identified by the differential method.**
**But, it happens that the reference images are not absolutely fully cloud-free, like this fig 5a, and 6b.**
**This is inherent of ELIFAN : once in the library, the image is not processed by the absolute process, since it has already been evaluated as a fully clear-sky image (see the organigramme of Fig.4 in the manuscript).**

6. Page 12, line 20: it is not clear if the method is dependent for solar zenith angle. The images in figure 3 correspond almost to the same solar zenith angle. What about a set of images at a lower solar zenith angle? Are the same thresholds valid?
**→ Yes, the same criteria are valid along the day. We have made this clearer in the text page 11 line 19.**

Moreover, it is very common that the RBR values change among different cameras. Apart from the

distribution of RBR, which is the effect on RBR threshold values?

**→ This is correct. One needs to adapt the thresholds to the different cameras (see Table 3), due to this effect.**

**For some systems that would generate images that are significantly different from an EKO or RAPACE system, it might be more difficult to find appropriate thresholds (systems with more contribution of the red color for example). In that case (not encountered here), one possibility may be to first apply a filter to the image in order to adapt the RBR pdf back into a similar range and variability than other cameras.**

7. Page 12, lines 22-23: Why do you choose now the pixel by pixel process but not in the previous step?

**→An image is first evaluated at step 3 from its RBR pdf : we evaluate whether it is a fully cloudy or a fully cloud-free image. Only afterward, is the image evaluated on a pixel-by-pixel point of view, by the differential or absolute processes. Those processes are appropriate for images that have a mix of cloudy and clear-sky pixels.**

**That was initially explained in the manuscript. See page 10, lines 2-4 : « In step 3, the principle is to consider the cropped image as a whole, and detect whether it is a fully cloudy sky image, a fully cloud free image, or neither of those two (i. e. a partly cloudy image). This step is based on the RBR distribution of the entire ensemble of pixels. » It is also clear from the organigramme in Figure 4.**

In this way, some cloudy areas (e.g. in figure 5a) could be removed from the "cloud-free" image. 8.

Figure 6 and relevant text: The sky image in figure 6b has been detected as partly cloudy.

**→ Yes, this happens about 17% of the time (outside the sunrise/sunset period). Those failing library images show only very small and/or thin clouds, which can have an impact of a few % in the final cloud cover estimate (no impact in the case shown though). It is an illustration of the limitation/weakness of the library. However, a discussion about this definitely lacked in the initial manuscript. We added some comments page 12, lines 15-17 ; page 16, lines 12-15.**

*(8)* However, it is almost identical with the image 5a.
Which is the pdf versus RBR for the 6b image?

**→ The pdf of Fig. 6b is shown below (left panel), compared to the former Fig.5a (middle panel) and the new Fig.5a (right panel).**

**One can see how a fully cloud-free image has only one mode in the pdf, toward the small values of the RBR. When there are some small clouds, a secondary small mode may appear, which is not obvious in the case of Fig. 6b.**

[Figure]

[Figure]

[Figure]

RBR pdf of Fig. 6b          RBR pdf of Former Fig. 5a          RBR pdf of New Fig. 5a

This example questions strongly the robustness of the cloud-free image selection method.

**→ We agree that the cloud-free images selection method is not perfect. Those examples illustrate the limitation of the library. We also have clearly exposed this limitation in section 3.4 and Table 4. 17 % of the library images have such a thin and small (or tiny) cloud in the field, rather than being perfectly cloud-free. But those correspond to a maximum of a few percent in the final amount of cloud cover, which remains acceptable for our purpose. We added more discussion in the final conclusion about this aspect.**

9. Table 3 and relevant text: the EKO camera at P2OA presents different Tclear and Tcloudy values when compared to those at SIRTA and CO-PDD.
Please explain the differences taking into account that the cropped radii are different.

**→ The different cropped radii comes from the different size image (Table 1). The twice bigger image of the P2OA EKO camera leads to a twice bigger radius of the cropped image. But this has no influence on the R/B thresholds.**

**This is more likely due to the altitude of the P2OA EKO (2877 m asl), associated with less aerosols, which shifts the cloud-free image pdf toward smaller RBR values.**
**We have added this comment on this in the revised version, page 14, lines 17-19.**

10. Page 17, line 20: the hours between 9 and 15UT correspond to which solar zenith angles in summer and winter time?

**→ The hours between 0900 and 1500 UTC correspond to minimum and maximum zenith angle of 20° and 42° respectively in Summer, and 66° and 78° respectively in winter.**

How this is related to the cropped sky area?

**→ The consequence of the summer/winter difference here is that the sun mask is entirely included in the cropped image of summer time (it corresponds to 10 % of the cropped image), but only 30 % (0900 or 1500 UTC) to 60 % (1200 UTC) of it is included in the cropped image during the smallest day of winter (which corresponds to 3 % to 6 % of the processed image respectively).**
**But this should not have a strong impact on the CF_diff versus CF_abs comparison.**

**We have added a comment on this aspect in the revised version page 18, lines 6-8.**

11. Section 4: It is good to know that an all-sky imager could provide useful information for cloudiness at any measuring site. It also explains much of the measured variability of surface irradiance. This seems to be a nice paragraph promoting all-sky imagers but it does not enhance the validity of the algorithm.

Moreover, the fact that an all-sky imager, compared to a ceilometer, could provide a holistic information for cloud cover (but not for cloud height) is totally expected.

**→ We fully agree with this.**

It is not clear what is the purpose of section 4, it may be removed.

**→ This question was also raised by Rev #2.**
**The main goal was to illustrate the algorithm process presented before, and discuss it with other points of view brought by complementary instruments.**
**Which eventually gives 2 objectives :**
**(1) illustrating the ELIFAN process and weakness/strength points presented before**
**(2) showing the complementarity of the instruments.**
**We thought that this section would be useful for readers who are interested in the application of the sky cameras.**

**But, based on the fact that :**
**- both Reviewer #1 and Reviewer #2 have put this section into question,**
**- we have inserted more illustrations of the strength and limitations of ELIFAN in section 3,**
**- it is appropriate to keep the manuscript to a reasonable length,**
**we propose to remove this section.**
**With this change, the previous evaluation section 3.4 has now been renamed section 4 itself, in order to better balance the size of the various parts of the revised article.**

**However, we are still willing to keep the original section 4 if ever the Reviewers and Editor finally find it more appropriate.**

12. Based on the above comments, the abstract and conclusion paragraphs should be revised.
**→ We have revised the abstract and conclusion accordingly.**

Minor comments: Page 9, line 4: the solar mask (instead of maks)
Figure 3: please align horizontally the 2 images
Figure 5: please check again the y axis title in figures d,e,f.

**We made those changes as suggested.**

---

## Author Response (AR2)

Manuscript #amt-2018-452
Reply to Editor, 2019/08/26
* * *
Dear Editor,

We thank you for your evaluation and recommandations. We also thank the reviewers for their analysis and advises.

We have addressed three points, covering the Editor and Reviewer minor revision recommandations :

From the two recommandations of the Editor :

1. We have completed the discussions on background studies, in order to include earlier works and present some of the historical context. Especially, we have cited Shields et al (2013), with complementary references of historical works.
Consistently with the structure of the paper though, we included those discussions at different places of the manuscript : in the Introduction for the historical context of sky camera applications, in section 2.2 about the different technologies, and in section 3.1 / table 2 when considering the different cloud cover retrieval techniques. Note that we restricted our background analysis in 3.1 to visible sky cameras, as initially.

*Corresponding modifications :*
 *Introduction, Page 2, lines 5-8 : added references to historical references, and reviews.*
 *Section 2.2, Page 4, lines 10-17, 24-27, 29 : Added discussions on historical developements.*
 *Section 3.1, Page 6, line 8 : Added references.*
 *Section 3.1, Page 7, lines 4-5 and Table 2 :Added references.*

2. For the second point of the editor (and of Reviewer #2), we have added a discussion on our RBR thresholds relatively to the link between the AOD and the cloud-free RBR determined by Ghonima et al 2012 with a TSI camera. We also included references to climatological studies of observed AOD over the world.
Since this discussion was quite long, we did not find appropriate to include it in the conclusion. Instead, we found this discussion to fit better into section 3.3, about the adaptation of the algorithm to different sky cameras. We have revised the Conclusion accordingly.

*Corresponding modifications :*
 *Section 3.3, Page 15-line 23 to Page 16-line 15 : Added discussion.*
 *Conclusion, Page 20, line 20-21 : Revised comment on prospectives, and simplification of the conclusion.*

Finally, we considered the first point of Reviewer #2 by adding a sentence at the beginning of section 3.2, which clearly states the limited innovation of ELIFAN.

*Corresponding modifications :*
 *Section 3.2, Page 8, lines 3-4 : Added comment.*

Point 2 of Reviewer#2 joins point 2 of the Editor.

With those three points, we hope that we have covered all recommandations.